# Stochastic Context Consistency Reasoning for Domain Adaptive Object Detection

## ABSTRACT

Domain Adaptive Object Detection (DAOD) aims to improve the adaptation of the detector for the unlabeled target domain by the labeled source domain. Recent advances leverage a self-training framework to enable a student model to learn the target domain knowledge using pseudo labels generated by a teacher model. Despite great successes, such category-level consistency supervision suffers from poor quality of pseudo labels. To mitigate the problem, we propose a **sto**chastic **c**ontext **c**onsist**e**ncy **r**easoning (SOC-CER) network with the self-training framework. Firstly, we introduce a stochastic complementary masking module (SCM) to generate complementary masked images thus preventing the network from over-relying on specific visual clues. Secondly, we design an inter-changeable context consistency reasoning module (Inter-CCR), which constructs an inter-context consistency paradigm to capture the texture and contour details in the target domain by aligning the predictions of the student model for complementary masked images. Meanwhile, we develop an intra-changeable context consistency reasoning module (Intra-CCR), which constructs an intra-context consistency paradigm to strengthen the utilization of context relations by utilizing pseudo labels to supervise the predictions of the student model. Experimental results on three DAOD benchmarks demonstrate our method outperforms current state-of-the-art methods by a large margin. *Code is released in supplementary materials.*

## CCS CONCEPTS

• **Computing methodologies** → **Object detection**.

## KEYWORDS

Domain adaptation, object detection, context consistency learning.

## 1 INTRODUCTION

Object detection plays a crucial role in applications such as autonomous driving, intelligent surveillance, and industrial automation. However, the detector trained on the curated dataset (*i.e.*, labeled source domain) suffers from severe adaptation degeneration in practical application environments (*i.e.*, unlabeled target domain). This is caused by the domain shift due to the discrepancy in the appearance and texture of objects, and the background. To address this problem, researchers turn their attention to Domain

*ACM MM, 2024, Melbourne, Australia*

© 2024 Copyright held by the owner/author(s). Publication rights licensed to ACM.
ACM ISBN 978-x-xxxx-xxxx-x/YY/MM
https://doi.org/10.1145/nnnnnnn.nnnnnnn

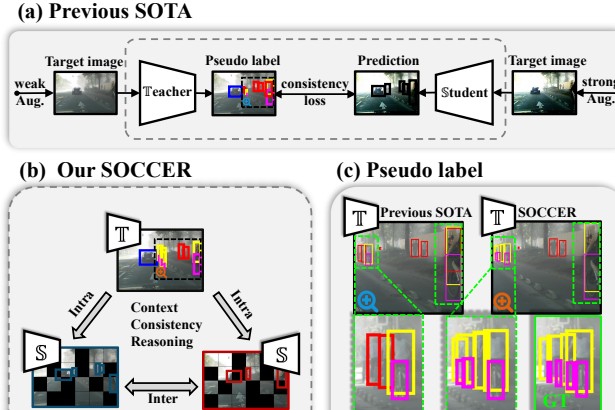

**(a) Previous SOTA**

**(b) Our SOCCER**     **(c) Pseudo label**

**Figure 1: (a) Previous self-training DAOD methods heavily rely on the pseudo label for consistency training. (b) SOCCER constructs the inter/intra-context consistency from multi-view by Inter-CCR and Intra-CCR modules, which enhance the representation and contextual correlation ability of the network. (c) The example of our pseudo label quality comparison with MIC [23]. Benefiting from Inter-CCR and Intra-CCR modules, the network extracts more discriminative features to distinguish confusing categories (*i.e.*, "rider" and "person") and utilizes more context clues to build relations between objects (*i.e.*, "rider" and "bicycle").**

Adaptive Object Detection (DAOD), intending to design a detector that can transfer the knowledge learned from the labeled source domain to the unlabeled target domain [6, 34, 63].

Unlike traditional object detection, the critical challenge of DAOD is to learn the domain-invariant feature for bridging the domain gap while preserving the domain-specific characteristic of the target domain for facilitating detection. The mainstream approaches try to align feature distributions between source and target domains through pixel-level [24, 30, 63], instance-level [18, 45, 63], and image-level [6, 31] adversarial learning. Other methods attempt to mitigate domain shifts by modeling prototypes as category centers to minimize each distance [48, 53, 59], using graph matching theory [13, 32, 34, 38, 56], and employing image-to-image translation to generate target-like images [9, 21, 44, 57]. However, the above methods focus on extracting domain-invariant features, and this might ignore domain-specific features that reflect the discriminatory information of objects in the target domain. Additionally, they neglect the context correlation of 1) between objects and 2) between objects and the background, which can provide powerful discriminatory clues for object detection, particularly under a large domain gap.

Recently, some DAOD methods [9, 21, 23, 27, 39, 50] introduce the self-training framework and achieve significant adaptation gains. The teacher model predicts the unlabeled target images to obtain pseudo labels so as to guide the student model to generate target features near the support of the source domain. Further, AT [36] adopts adversarial learning to mitigate the domain shifts toward the source domain in the student model. CMT [2] integrates contrastive learning to fully exploit the ability of the teacher model to characterize the knowledge of the target domain. The existing self-training frameworks heavily rely on pseudo labels of the teacher model to implement the category-level consistency constraint, as illustrated in Figure 1 (a). However, the quality of pseudo labels is usually poor, and such single category-level consistency constraint limits the domain adaptation ability of the student model thus leading to sub-optimal learning.

To address the above problems, we propose a **sto**chastic **c**ontext **c**onsistency **r**easoning (SOCCER) network to learn the context correlation knowledge of the target domain by the multi-view context consistency (as shown in Figure 1 (b)). Firstly, we introduce a stochastic complementary masking module (SCM) to generate a pair of complementary interchangeable views by stochastically masking a part of local visual clues. This complementary masking strategy prevents the student model from taking shortcuts with overlapping visual clues to truly learn the relevance of neighboring contextual regions. Secondly, we design an inter-changeable context consistency reasoning module (Inter-CCR) to model the inter-context consistency between the student model's predictions from interchangeable views. Enforcing inter-context consistency between two views enhances bidirectional feature alignment and improves the representation ability of extracted features for texture and appearance attributes. Meanwhile, we develop an interchangeable context consistency reasoning module (Intra-CCR) to impose intra-context consistency for multi-view between pseudo-labels generated by the teacher model with a complete view and predictions of the student model for incomplete views. This fully strengthens the potential of the network for contextual correlation, thereby improving both the cross-domain adaptability and the quality of pseudo-labels, as shown in Figure 1 (c). Extensive experiments on three cross-domain benchmarks demonstrate the extraordinary adaptation ability of SOCCER and the effectiveness of each component.

The contributions of this paper are summarized as follows:

- We propose a stochastic context consistency reasoning network to learn the context correlation knowledge of the target domain for modeling the underlying contextual relationship of the target domain between objects and environments.
- We introduce stochastic complementary masking to exploit the discriminate visual clues in the target domain, thus preventing the network from over-relying on specific regions of objects.
- We design Inter-CCR and Intra-CCR modules to construct two kinds of inter/intra-context consistency, thereby enhancing feature representation, modeling contextual correlation, and improving the quality of pseudo labels.
- Extensive experiments demonstrate that SOCCER achieves state-of-the-art detection performance on three benchmarks and outperforms existing approaches significantly.

## 2 RELATED WORK

**Object detection.** Object detection can be divided into two categories: one-stage (*e.g.*, YOLO [40], FCOS [47] and Deformable DETR [64]) and two-stage detectors (*e.g.*, Faster RCNN [41] and Fast RCNN [14]). One-stage detectors directly predict object bounding boxes and their associated class probabilities for the entire image without the need for explicit region proposals. The two-stage detectors first generate some candidate proposals by the region proposal network (RPN) [41] and then refine these candidates to give the final bounding boxes and categories. We employ Faster RCNN as the base detector for its outstanding performance and expansibility.
**Domain adaptive object detection.** DAOD aims to train a detector with labeled source domain data and unlabeled target domain data, in order to achieve satisfactory adaptive performance on the target domain. The earliest DAOD is based on adversarial feature learning. Domain Adaptive Faster RCNN [6] applies domain adaptation to object detection, which proposes adversarial-based feature alignment for Faster RCNN at image and instance level to mitigate the domain shift. It leads a series of methods [3, 7, 22, 30, 52, 63] to explore different aspects of feature alignment. For example, context-aware alignment [3], multi-level alignment [22, 24, 52, 63], token/query-based alignment [50, 56],and multi-scale alignment [7]. Recently, the idea of Mean Teacher [46] is extended from semi-supervised object detection to DAOD by MTOR [1] and achieves many remarkable works [10, 23, 29, 36, 39, 56].
**Consistency learning for DAOD.** Consistency learning for DAOD aims to train a robust detector by making similar predictions about different perturbations. Jeong *et al.* [26] propose CSD to constrain the prediction consistency of the model by feeding the original image and the horizontally flipped image. Xie *et al.* [51] propose a pretext task for pixel-level consistency. Recently, researchers combine consistency learning with the self-training framework to address domain adaptation in object detection. Kennerley *et al.* [29] introduce a two-phase consistency network to enhance the consistency of prediction between the teacher and the student model. Hoyer *et al.* [23] propose MIC to strengthen the reasoning ability for context by constraining the consistency of the prediction between the original image and the mask image. Different from the above approaches: firstly, our method models both teacher-student and student-student consistency constraints, while previous works only focus on the consistency of the teacher-student aspect and thus heavily rely on pseudo labels. Secondly, our proposed context consistency reasoning enforces constraints from both classification and localization perspectives, ensuring the model learns both contextual semantic information and spatial information.

## 3 PRELIMINARIES

### 3.1 Problem Formulation

Given a labeled source domain $\mathcal{D}_s = (X_s, B_s, C_s)$ and an unlabeled target domain $\mathcal{D}_t = (X_t)$, where $X_s$ and $X_t$ present $N_s$ source images and $N_t$ target images, $B_s = \{b_s^i\}_{i=1}^{N_s}$ denotes the bounding box annotations and $C_s = \{c_s^i\}_{i=1}^{N_s}$ denotes corresponding class labels. We train a domain adaptive detector with $\mathcal{D}_s$ and $\mathcal{D}_t$, and the ultimate goal is to design a detector that performs effectively within the target domain.

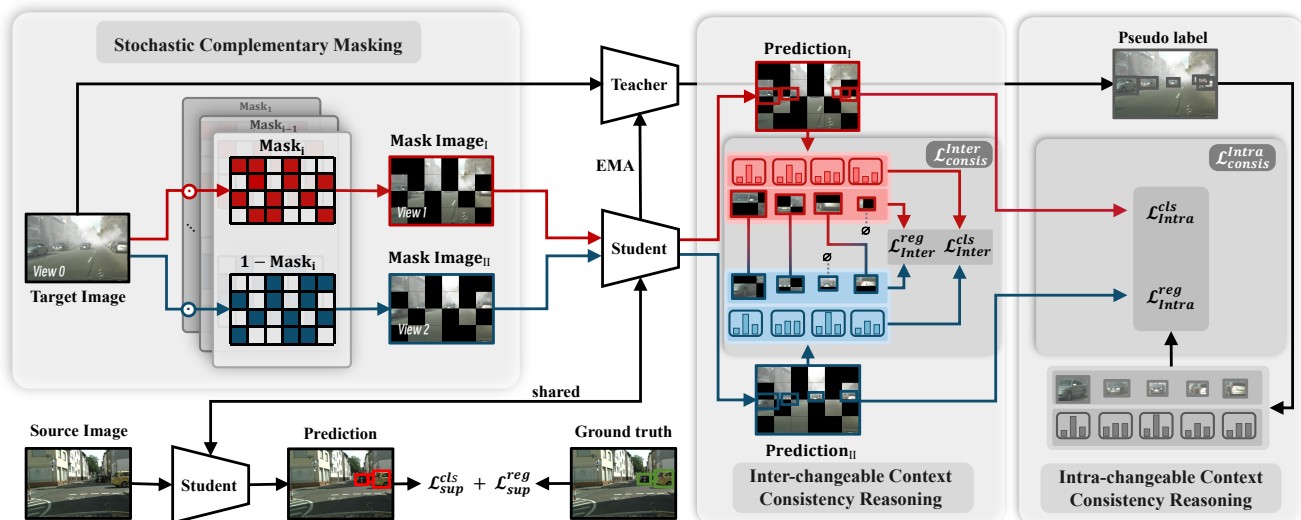

Figure 2: Overview of the proposed SOCCER, which comprises three modules: 1) Stochastic complementary masking (SCM) generates pairs of interchangeable views of target images. 2) Inter-changeable context consistency reasoning (Inter-CCR) models the inter-context consistency by aligning predictions of the student model for two views to enhance the representation of features. 3) Intra-changeable context consistency reasoning (Intra-CCR) conducts intra-context consistency between pseudo labels and student model's predictions from interchangeable views to learn contextual semantic and spatial information.

## 3.2 Self-training Framework

We use SADA [7] as the student model $\mathbb{S}$, which is a two-stage Faster RCNN [41] object detector combined with image-level and instance-level adversarial learning. Domain discriminators $\mathbb{D}$ are placed after the student module to predict the domain label of the features. The adversarial optimization function is formulated:

$$\mathcal{L}_{adv} = \max_{\mathbb{S}} \min_{\mathbb{D}} \mathcal{L}_{BCE}, \qquad (1)$$

where $\mathcal{L}_{BCE}$ denotes the BCE loss of domain classification at the image/instance-level and we use Gradient Reverse Layers(GRL) [12] for min-max optimization.

The teacher model and the student model share the same structure in the self-training framework. The student model acquires the source domain knowledge from source data by supervised training:

$$\mathcal{L}_{sup}(X_s, B_s, C_s) = \mathcal{L}_{sup}^{cls}(C_s', C_s) + \mathcal{L}_{sup}^{reg}(B_s', B_s), \qquad (2)$$

where $C_s'$ and $B_s'$ denote categories and bounding boxes in $X_s$ predicted by the student model. By forcing the student model to imitate the deep representations of the teacher model, the student model can learn the knowledge of the target domain from the pseudo labels $(\hat{B}, \hat{C})$:

$$\mathcal{L}_{unsup}(X_t, \hat{B}, \hat{C}) = \mathcal{L}_{unsup}^{cls}(C_t', \hat{C}) + \mathcal{L}_{unsup}^{reg}(B_t', \hat{B}), \qquad (3)$$

where $C_t'$ and $B_t'$ denote categories and bounding boxes in $X_t$ predicted by the student model, $\hat{C}$ and $\hat{B}$ denote the categories and bounding boxes of objects in a pseudo label. In addition, to filter out the noisy pseudo labels generated by the teacher model, we apply a confidence threshold $\delta$ to filter low-confidence bounding boxes and remove duplicate bounding boxes using non-maximum

suppression (NMS) [15] for each class. The teacher model is updated by Exponential Moving Average (EMA) from the weights of the student model without gradient accumulation:

$$\theta_t \leftarrow \alpha\theta_t + (1 - \alpha)\theta_s, \qquad (4)$$

where $\theta_t$ and $\theta_s$ denote the model parameters of teacher and student respectively, and $\alpha$ is an update hyper-parameter.

## 4 METHODOLOGY

### 4.1 Overview

We propose the SOCCER network for DAOD, which is composed of SCM, Inter-CCR, and Intra-CCR, as illustrated in Figure 2. SOCCER is built on the self-training framework as introduced in Section 3.2. Concretely, SCM generates a pair of interchangeable views $\{\mathcal{V}_1, \mathcal{V}_2\}$ by stochastically masking a part of local visual features in target images. Then, Inter-CCR is designed to conduct consistency reasoning between predictions of $\{\mathcal{V}_1, \mathcal{V}_2\}$ from the student model, thus representing more domain-specific clues (i.e., appearance, texture). Simultaneously, in Intra-CCR, the teacher model generates pseudo labels from the original view $\mathcal{V}_0$ (target image $X_t$) with complete context information and supervises the reasoning of the student model for interchangeable views $\{\mathcal{V}_1, \mathcal{V}_2\}$ with the incomplete complementary context information. Therefore, both teacher and student models in this network acquire a powerful ability of contextual correlation.

### 4.2 Stochastic Complementary Masking

To enhance the context correlation ability of the network on the target domain, we employ stochastic complementary mask pairs

$\{\mathcal{M}, \overline{\mathcal{M}}\}$ to mask patches of the target image stochastically and generate interchangeable views $\{\mathcal{V}_1, \mathcal{V}_2\}$. The network is encouraged to utilize local clues in remaining visible regions to reason masked regions. Concretely, a stochastic mask $\mathcal{M}$ is generated by:

$$\mathcal{M}_{hb:(h+1)b, wb:(w+1)b} = \begin{cases} 1, & \text{if } u > r \\ 0, & \text{otherwise} \end{cases}, \qquad (5)$$

where $u$ is sampled from a uniform distribution $\mathcal{U}(0,1)$, $b$ is the spatial size of a masked patch, $r$ is the mask ratio in $(0,1)$, $h \in [0, \ldots, \frac{H}{b} - 1]$, $w \in [0, \ldots, \frac{W}{b} - 1]$ are the masked patch indices, and H, W are the height and width of the input image. Then, the complementary mask $\overline{\mathcal{M}}$ is obtained as:

$$\overline{\mathcal{M}} = \mathbf{1} - \mathcal{M}, \qquad (6)$$

where $\mathbf{1}$ represents a all-ones matrix with the same dimensions as $\mathcal{M}$. The interchangeable target views are obtained by element-wise multiplication between the complementary masks and the target domain images:

$$\begin{aligned} \mathcal{V}_1 &= \mathcal{M} \odot X_t, \\ \mathcal{V}_2 &= \overline{\mathcal{M}} \odot X_t. \end{aligned} \qquad (7)$$

SCM explicitly constructs views $\{\mathcal{V}_1, \mathcal{V}_2\}$ with inter-context relationships, and forms intra-context relationships with the original view $\mathcal{V}_0$. It also achieves a better balance between deletion and preservation of regional information in images compared to other methods [5, 11, 61].

## 4.3 Inter-changeable Context Consistency Reasoning

Previous self-training methods mainly construct the category-level consistency constraint but suffer from the poor quality of pseudo labels. To mitigate the reliance on pseudo-labels, we design the Inter-CCR module, which leverages self-supervision signals from the student model to bidirectionally align features, thereby enhancing features' representation capability without the supervision of pseudo-labels. Specifically, the interchangeable views $\{\mathcal{V}_1, \mathcal{V}_2\}$ are first fed into the student model to generate corresponding predictions. Subsequently, the classification consistency reasoning $\mathcal{L}_{Inter}^{cls}$ and regression consistency reasoning $\mathcal{L}_{Inter}^{reg}$ are conducted to the predictions of interchangeable views. To avoid mismatched consistency reasoning, we set a large IoU matching threshold $\tau$ as 0.75. Here, we calculate the IoU matrix for $B_t^{\mathcal{M}}$ and $B_t^{\overline{\mathcal{M}}}$, and select the maximum value along the longest side as their matching score. If this score exceeds the $\tau$, the bounding boxes are considered successfully matched. More details in Section 5.4 and Appendix.

For the classification consistency reasoning, we bidirectionally align the category distribution between category probability vectors $P^{\mathcal{M}}$ and $P^{\overline{\mathcal{M}}}$ of interchangeable views $\{\mathcal{V}_1, \mathcal{V}_2\}$:

$$\mathcal{L}_{Inter}^{cls}(P^{\mathcal{M}}, P^{\overline{\mathcal{M}}}) = -\frac{1}{2K} \sum_{i=1}^{K} \sum_{j=1}^{K} [w_j c_j^{\overline{\mathcal{M}}} log(p_i^{\mathcal{M}}) + w_i c_i^{\mathcal{M}} log(p_j^{\overline{\mathcal{M}}})], \qquad (8)$$

where $K$ is the number of matched bounding box pairs between $B_t^{\mathcal{M}}$ and $B_t^{\overline{\mathcal{M}}}$, $p_i^{\mathcal{M}}$ is the category probability vector of the $i^{th}$ bounding

box in $B_t^{\mathcal{M}}$, $c_j^{\overline{\mathcal{M}}}$ denotes the corresponding category prediction in $B_t^{\overline{\mathcal{M}}}$, and $w_j$ is the confidence of $c_j^{\overline{\mathcal{M}}}$ in $(0,1)$. For the regression consistency reasoning, we introduce the Huber loss to bidirectionally align the matched bounding boxes in $B_t^{\mathcal{M}}$ and $B_t^{\overline{\mathcal{M}}}$:

$$\mathcal{L}_{Inter}^{reg}(B_t^{\mathcal{M}}, B_t^{\overline{\mathcal{M}}}) = -\frac{1}{K} \sum_{n=1}^{K} H_\sigma(b_n^{\mathcal{M}} - b_n^{\overline{\mathcal{M}}}), \qquad (9)$$

where $B_t^{\mathcal{M}}, B_t^{\overline{\mathcal{M}}}$ denote the predicted bounding boxes and the Huber loss is defined as:

$$H_\sigma(x) = \begin{cases} \frac{1}{2}(x)^2, & \text{if } |x| \le \sigma \\ \sigma|x| - \frac{1}{2}\sigma^2, & \text{otherwise} \end{cases}, \qquad (10)$$

to select a more appropriate transition point $\sigma$, instead of using a hard threshold, we dynamically adjust $\sigma$, which is set to the average difference between $b_n^{\mathcal{M}}$ and $b_n^{\overline{\mathcal{M}}}$ during training, and locks below a certain value. It reduces the sensitivity of the model to outliers during initial training and adaptively increases the punishment. More details are discussed in Section 5.4.

The total Inter-CCR loss is defined as:

$$\mathcal{L}_{consis}^{Inter} = \mathcal{L}_{Inter}^{cls} + \mathcal{L}_{Inter}^{reg}. \qquad (11)$$

## 4.4 Intra-changeable Context Consistency Reasoning

We introduce the Intra-CCR to construct a self-training framework with intra-context consistency, in which the teacher model encourages the student model to explore extra target contextual correlation thus reasoning the masked regions. During the teacher-student mutual learning phase, this contextual correlation ability can be updated to the teacher model through EMA, thereby improving the quality of pseudo-labels and creating a virtuous cycle. Concretely, the teacher model obtains the complete information of the original view $\mathcal{V}_0$ to generate pseudo labels $(\hat{B}, \hat{C})$. Afterward, we utilize pseudo labels to supervise predictions of the student model from $\{\mathcal{V}_1, \mathcal{V}_2\}$ by classification and regression context consistency reasoning losses: $\mathcal{L}_{Intra}^{cls}$ and $\mathcal{L}_{Intra}^{reg}$.

The classification consistency reasoning optimizes the student model to reason contextual semantic information from masked views with:

$$\mathcal{L}_{Intra}^{cls}(P^{\mathcal{M}}, P^{\overline{\mathcal{M}}}, \hat{C}) = -[\frac{\lambda}{N} \sum_{i=1}^{N} \hat{w}_i \hat{c}_i log(p_i^{\mathcal{M}}) \\ + \frac{\mu}{M} \sum_{j=1}^{M} \hat{w}_j \hat{c}_j log(p_j^{\overline{\mathcal{M}}})], \qquad (12)$$

where $N$ and $M$ represent the number of objects detected by the student model while reasoning $\mathcal{V}_1$ and $\mathcal{V}_2$, $\hat{c}_i$ denotes the corresponding category prediction in the $i^{th}$ bounding box $B_t^{\mathcal{M}}$, and $\hat{w}_i = max(softmax(p_i^{\mathcal{M}}))$ is the confidence of $\hat{c}_i$. As the actual number of objects in $\{\mathcal{V}_1, \mathcal{V}_2\}$ is different, we set $\lambda = \frac{N}{N+M}$, $\mu = \frac{M}{N+M}$ to balance the contributions of two loss branches of Intra-CCR. We also discuss another weight setting in Appendix. Meanwhile, the regression consistency reasoning enhances the space perception capability of the student by reasoning the location of masked objects

in $\{\mathcal{V}_1, \mathcal{V}_2\}$:

$$\mathcal{L}_{Intra}^{reg}(B_t^{\mathcal{M}}, B_t^{\overline{\mathcal{M}}}, \hat{B}) = -[\frac{\lambda}{N}\sum_{i=1}^{N}H_\sigma(b_i^{\mathcal{M}} - \hat{b}_i) \qquad (13)$$
$$+ \frac{\mu}{M}\sum_{j=1}^{M}H_\sigma(b_j^{\overline{\mathcal{M}}} - \hat{b}_j)] .$$

The total Intra-CCR loss is defined as:

$$\mathcal{L}_{consis}^{Intra} = \mathcal{L}_{Intra}^{cls} + \mathcal{L}_{Intra}^{reg} . \qquad (14)$$

### 4.5 Overall Optimization Objective

The overall objective of SOCCER is:

$$\mathcal{L} = \lambda_0 \cdot \mathcal{L}_{sup} + \lambda_1 \cdot \mathcal{L}_{consis}^{Inter} + \lambda_2 \cdot \mathcal{L}_{consis}^{Intra} + \lambda_3 \cdot \mathcal{L}_{adv}, \qquad (15)$$

where $\mathcal{L}_{sup}$ is the supervised loss from Eq. 2, $\mathcal{L}_{consis}^{Inter}$ is Inter-CCR loss, $\mathcal{L}_{consis}^{Intra}$ is Intra-CCR loss, and $\mathcal{L}_{adv}$ is adversarial loss given by Eq. 1, $\lambda_{0-4}$ are weights of different loss. We present the detailed pseudo-code of the training pipeline in Appendix.

## 5 EXPERIMENTS

### 5.1 Datasets

We use the mean Average Precision (mAP) with a threshold of 0.5 to evaluate the detection performance on target domains and evaluate the effectiveness of the SOCCER on four public datasets, including BDD100k [55] (**B**), Sim10k [28] (**S**), Cityscapes [8] (**C**), and Foggy Cityscapes [42] (**F**).

**Cityscapes to BDD100k:** BDD100k [55] is a large-scale driving dataset. Following methods [10, 20, 35, 52], we extract the daytime subset of BDD100k as the target domain, containing 36, 728 training images and 5, 258 validation images. The subset of Cityscapes with seven shared categories is adopted as the source domain.

**Sim10k to Cityscapes:** Sim10k [28] is a synthetic dataset from GTA-V game engine containing 10,000 images with 58,071 annotations of the car. We use Sim10k as the source domain and a subset of Cityscapes with only the "car" category as the target domain.

**Cityscapes to Foggy Cityscapes:** Cityscapes [8] is collected from 50 urban scenes of normal weather and contains 2,975 images for training and 500 images for validation. Foggy Cityscapes [42] is synthesized from the Cityscapes to simulate foggy weather. We use Cityscapes as the source domain and Foggy Cityscapes with the most dense fog (0.02) as the target domain.

### 5.2 Implementation Details

For a fair comparison with SOTA methods, we adopt the Faster RCNN object detector [41] with the ResNet-50 backbone [19] and FPN [37] as the detection model. We scale all images by resizing the shorter side of the image to 800 pixels while maintaining the image ratios. We set the size of masked patches $b = 32$, the mask ratio $r = 0.5$, the EMA update ratio $\alpha = 0.9$, and the confidence threshold $\delta = 0.8$. We copy the weight of the student model to the teacher model from the start without a burn-in stage, and mutual learning for 60k iterations with an initial learning rate of 0.0025, and a weight decay of 0.0001. All experiments are implemented using Detectron [16] and conducted on a single RTX 4090 GPU with a batch size of 2 (*i.e.*, 1 source image and 1 target image).

### 5.3 Comparison with SOTA

In this section, we compare the SOCCER with other SOTA in three DAOD benchmarks, including Cityscapes to BDD100k ( Table 1), Sim10k to Cityscapes (Table 2), and Cityscapes to Foggy Cityscapes (Table 3). "Source" denotes the base Faster RCNN only trained by source data as the performance lower bound.

**Cityscapes to BDD100k:** In Table 1, SOCCER achieves 41.8% mAP, surpassing all comparison models by an average significant margin of 6.4% mAP over the best-performing Faster RCNN detector MIC [23], the FCOS detector SIGMA [34], and the Deformable DETR detector MTM [50]. And SOCCER achieves the best performance in almost all categories, especially in rare categories such as "truck", "mcycle" and "bicycle", as well as confusing categories like "person" and "rider", which are also reflected in Figure 7. Benefiting from the Inter/Intra-CCR to enforce context consistency between multi-view, SOCCER gains an average 7.2% mAP than other self-training methods, such as PT [4], MIC, and MTTrans [56].

**Sim10k to Cityscapes:** Table 2 validates the adaptation ability of SOCCER from synthetic to natural images with a large domain gap. SOCCER achieves 63.8% AP, outperforming the second-best method OADA [54] by 4.6% AP. We also notice that SOCCER exceeds the SOTA consistency learning based method MIC with 4.9% AP, which demonstrates the advantage of the stochastic complementary masking strategy.

**Cityscapes to Foggy Cityscapes:** In Table 3, SOCCER achieves 51.1% mAP, the best performance of all other SOTA. The comparison with the latest self-training methods CMT [2], MIC, and MTM demonstrates the effectiveness of SOCCER. The detection results of "car", "rider", "mcycle", and "bicycle" is higher than other methods, which demonstrates that SOCCER enhances the ability of context reasoning in correlated categories (as exampled in the detection results of "rider" and "bicycle" in Figure 1 (c) and Figure 5 $row1$ ) and intensive categories (as shown in Figure 5 $row2$).

### 5.4 Ablation Studies and Analysis

**Ablation Studies:** We conduct experiments on different variants of our model to evaluate the effectiveness of each component, and all ablation studies are performed on three benchmarks, as shown in Table 4. Comparison between $row\ A$ (baseline) and $row\ B$ shows that self-training and masking strategy in SCM bring obvious improvement. With our SCM ($row\ C$), adopting the dual branches strategy can achieve an average gain of 0.4% mAP compared to the single branch ($row\ B$). Compared with $row\ C$, $\mathcal{L}_{Intra}^{reg}$ ($row\ D$) enhances the localization ability, obtaining an average gain of 2.0% mAP, especially a gain of 4.1% mAP on $\mathbf{C} \rightarrow \mathbf{B}$. With $\mathcal{L}_{Inter}^{reg}$ and $\mathcal{L}_{Inter}^{cls}$ ($row\ E$), the performance on average increase by 2.0% than $row\ C$, which proves the effectiveness of Inter-CCR. The variant in $row\ E$ is boosted by $\mathcal{L}_{Intra}^{reg}$ ($row\ H$) with a performance average gain of 2.2% mAP on three benchmarks, which indicates that Intra-CCR can benefit from localization ability. Further, we remove $\mathcal{L}_{Inter}^{reg}$ ($row\ F$) and $\mathcal{L}_{Inter}^{cls}$ ($row\ G$) separately resulting in a performance decline over all three benchmarks, which proves that the single type of context consistency reasoning cannot align features well, thus reducing representation ability of features for spatial and semantic information. Note that, since $\mathcal{L}_{Intra}^{cls}$ is the basic loss of the self-training framework, we don't remove this.

**Table 1: Results of Cityscapes to BDD100k (daytime).The average precision (AP, %) on all classes is presented. FRCNN denotes Faster RCNN and DefDETR denotes Deformable DETR.**

| Method | Venues | Detector | person | rider | car | truck | bus | mcycle | bicycle | mAP |
|---|---|---|---|---|---|---|---|---|---|---|
| Source [41] | NeurIPS'15 | FRCNN | 48.2 | 32.6 | 63.4 | 9.3 | 6.6 | 14.5 | 23.2 | 28.3 |
| SADA [7] | IJCV'21 | FRCNN | 40.5 | 30.8 | 65.1 | 16.8 | 18.3 | 14.1 | 25.1 | 30.1 |
| TDD [21] | CVPR'22 | FRCNN | 39.6 | 38.9 | 53.9 | 24.1 | 25.5 | 24.5 | 28.8 | 33.6 |
| PT [4] | ICML'22 | FRCNN | 40.5 | 39.9 | 52.7 | 25.8 | **33.8** | 23.0 | 28.8 | 34.9 |
| MIC [23] | CVPR'23 | FRCNN | 49.8 | 36.8 | 68.1 | 24.0 | 25.6 | 18.7 | 30.7 | 36.2 |
| EPM [24] | ECCV'20 | FCOS | 39.6 | 26.8 | 55.8 | 18.8 | 19.1 | 14.5 | 20.1 | 27.8 |
| SIGMA [34] | CVPR'22 | FCOS | 46.9 | 29.6 | 64.1 | 20.2 | 23.6 | 17.9 | 26.3 | 32.7 |
| SFA [49] | ACM MM'21 | DefDETR | 40.2 | 27.6 | 57.5 | 19.1 | 23.4 | 15.4 | 19.2 | 28.9 |
| AQT [25] | IJCAI'22 | DefDETR | 38.2 | 33.0 | 58.4 | 17.3 | 18.4 | 16.9 | 23.5 | 29.4 |
| O$^2$net [17] | ACM MM'22 | DefDETR | 40.4 | 31.2 | 58.6 | 20.4 | 25.0 | 14.9 | 22.7 | 30.5 |
| MTTrans [56] | ECCV'22 | DefDETR | 44.1 | 30.1 | 61.5 | 25.1 | 26.9 | 17.1 | 23.0 | 32.6 |
| BiADT [20] | ICCV'23 | DefDETR | 42.0 | 34.5 | 59.9 | 17.2 | 19.2 | 17.8 | 24.4 | 32.7 |
| MTM [50] | AAAI'24 | DefDETR | 53.7 | 35.1 | 68.8 | 23.0 | 28.8 | 23.8 | 28.0 | 37.3 |
| SOCCER(ours) | - | FRCNN | **56.8** | **42.2** | **73.1** | **31.0** | 29.5 | **26.1** | **33.8** | **41.8** |

**Table 2: Results of Sim10k to Cityscapes (car).**

| Method | Venues | Detector | AP$_{car}$ |
|---|---|---|---|
| Source [41] | NeurIPS'15 | FRCNN | 34.5 |
| SADA [7] | IJCV'21 | FRCNN | 55.8 |
| TDD [21] | CVPR'22 | FRCNN | 53.4 |
| MGA [63] | CVPR'22 | FRCNN | 54.6 |
| PT [4] | ICML'22 | FRCNN | 55.1 |
| SAD [62] | T-PAMI'23 | FRCNN | 49.2 |
| MIC [23] | CVPR'23 | FRCNN | 58.9 |
| SCAN [33] | AAAI'22 | FCOS | 52.6 |
| SIGMA [34] | CVPR'22 | FCOS | 53.7 |
| OADA[54] | ECCV'22 | FCOS | 59.2 |
| CSDA [13] | ICCV'23 | FCOS | 57.8 |
| IGG [32] | ACM MM'23 | FCOS | 58.4 |
| CIGAR [38] | CVPR'23 | FCOS | 58.5 |
| AQT [25] | IJCAI'22 | DefDETR | 53.4 |
| O$^2$net [17] | ACM MM'22 | DefDETR | 54.1 |
| MTTrans[56] | ECCV'22 | DefDETR | 57.9 |
| DA-DETR [58] | CVPR'23 | DefDETR | 54.7 |
| BiADT [20] | ICCV'23 | DefDETR | 55.8 |
| MTM [50] | AAAI'24 | DefDETR | 58.1 |
| SOCCER(ours) | - | FRCNN | **63.8** |

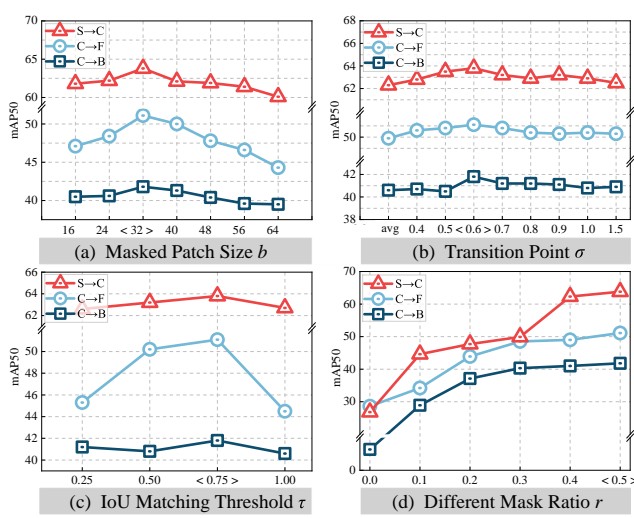

**Figure 3: Parameter sensitivity experiment of $b$, $\sigma$, $\tau$, and $r$. When adjusting one parameter, the other three parameters in $< \cdot >$ remain unchanged.**

**Parameter Sensitivity:**

**(1) Masked patch size:** Figure 3 (a) demonstrates the influence of different masked patch sizes $b$ for SOCCER. From the results, we find that the medium value (b=32 and b=40) achieves superior performance, and the too small and large patches with the same mask ratio ($r = 0.5$) achieve inferior results, especially in $C \rightarrow F$. The small patches can not mask sufficient regions, and the student model can get the correct detection result only by simple reasoning. In contrast, the large patches mask too many regions, even the whole object, thus the student model is difficult to reason the masked visual clues and gets caught in the sub-optimal solution.

**(2) Transition point:** In Figure 3 (b), we draw the results of SOCCER under different hyper-parameter $\sigma$ in Eq. 10. Note that when

$\sigma \geq 1$, it is fixed directly without dynamic adjustment, and $avg$ denotes the average difference of bounding boxes is not fixed. Overall, the mAP of all hyper-parameter settings floats within 1.5% and the best performance is achieved when $\sigma = 0.6$. We set $\sigma = 0.6$ for balancing the penalty strength.

**(3) IoU matching threshold:** In Figure 3 (c), we explore the effect of different matching thresholds $\tau$ in Inter-CCR on the performance. We set four groups $\tau$ on three benchmarks, and find that either too harsh or too loose matching thresholds are detrimental to Inter-CCR training. Too small $\tau$ leads to incorrect matches, while too large $\tau$ overlooks the intended matches. It is optimal only if $\tau = 0.75$ to avoid inconsistent matching objects.

**(4) Mask ratio:** In Figure 3 (d), we adjust the mask ratio $r$ in Eq. 5 to explore the influence of a pair of mask images with complementary mask degrees on SOCCER (i.e., when $\mathcal{V}_1$' mask ratio is $r$, $\mathcal{V}_2$' mask

**Table 3: Results of Cityscapes to Foggy Cityscapes (0.02, dense fog). The average precision (AP, %) on all classes is presented.**

| Method | Venues | Detector | person | rider | car | truck | bus | train | mcycle | bicycle | mAP |
|---|---|---|---|---|---|---|---|---|---|---|---|
| Source [41] | NeurIPS'15 | FRCNN | 39.1 | 22.1 | 42.2 | 20.1 | 30.0 | 6.6 | 28.5 | 35.4 | 30.2 |
| SADA [7] | IJCV'21 | FRCNN | 50.3 | 45.4 | 62.1 | 32.4 | 48.5 | 52.6 | 31.5 | 29.5 | 44.0 |
| TIA [60] | CVPR'22 | FRCNN | 34.8 | 46.3 | 49.7 | 31.1 | 52.1 | 48.6 | 37.7 | 38.1 | 42.3 |
| PT [4] | ICML'22 | FRCNN | 40.2 | 48.8 | 59.7 | 30.7 | 51.8 | 30.6 | 35.4 | 44.5 | 42.7 |
| TDD [21] | CVPR'22 | FRCNN | 39.6 | 47.5 | 55.7 | 33.8 | 47.6 | 42.1 | 37.0 | 41.4 | 43.1 |
| MGA [63] | CVPR'22 | FRCNN | 45.7 | 47.5 | 60.6 | 31.0 | 52.9 | 44.5 | 29.0 | 38.0 | 43.6 |
| SAD [62] | T-PAMI'23 | FRCNN | 38.3 | 47.2 | 58.8 | 34.9 | 57.7 | 48.3 | 35.7 | 42.0 | 45.2 |
| MIC [23] | CVPR'23 | FRCNN | **52.4** | 47.5 | 67.0 | **40.6** | 50.9 | **55.3** | 33.7 | 33.9 | 47.6 |
| CMT [2] | CVPR'23 | FRCNN | 45.9 | 55.7 | 63.7 | 39.6 | **66.0** | 38.8 | 41.4 | 51.2 | 50.3 |
| SCAN [33] | AAAI'22 | FCOS | 41.7 | 43.9 | 57.3 | 28.7 | 48.6 | 48.7 | 31.0 | 37.3 | 42.1 |
| SIGMA [34] | CVPR'22 | FCOS | 44.0 | 43.9 | 60.3 | 31.6 | 50.4 | 51.5 | 31.7 | 40.6 | 44.2 |
| OADA [54] | ECCV'22 | FCOS | 47.8 | 46.5 | 62.9 | 32.1 | 48.5 | 50.9 | 34.3 | 39.8 | 45.4 |
| CIGAR [38] | CVPR'23 | FCOS | 46.1 | 47.3 | 62.1 | 27.8 | 56.6 | 44.3 | 33.7 | 41.3 | 44.9 |
| CSDA [13] | ICCV'23 | FCOS | 46.6 | 46.3 | 63.1 | 28.1 | 56.3 | 53.7 | 33.1 | 39.1 | 45.8 |
| IGG [32] | ACM MM'23 | FCOS | 44.3 | 44.8 | 62.2 | 35.8 | 54.2 | 50.7 | 38.2 | 38.7 | 46.1 |
| MTTrans [56] | ECCV'22 | DefDETR | 47.7 | 49.9 | 65.2 | 25.8 | 45.9 | 33.8 | 32.6 | 46.5 | 43.4 |
| O²net [17] | ACM MM'22 | DefDETR | 48.7 | 51.5 | 63.6 | 31.1 | 47.6 | 47.8 | 38.0 | 45.9 | 46.8 |
| AQT [25] | IJCAI'22 | DefDETR | 49.3 | 52.3 | 64.4 | 27.7 | 53.7 | 46.5 | 36.0 | 46.4 | 47.1 |
| DA-DETR [58] | CVPR'23 | DefDETR | 49.9 | 50.0 | 63.1 | 24.0 | 45.8 | 37.5 | 31.6 | 46.3 | 43.5 |
| BiADT [20] | ICCV'23 | DefDETR | 50.7 | 56.3 | 67.1 | 28.8 | 53.7 | 49.5 | 38.8 | 50.1 | 49.4 |
| MTM [50] | AAAI'24 | DefDETR | 51.0 | 53.4 | 67.2 | 37.2 | 54.4 | 41.6 | 38.4 | 47.7 | 48.9 |
| SOCCER(ours) | - | FRCNN | 51.7 | **57.7** | **68.6** | 38.2 | 51.6 | 47.5 | **41.6** | **51.7** | **51.1** |

**Table 4: Ablation studies of SOCCER on three groups of domain adaptation experiments. We report mean average precision (mAP, %) on each of the combinations. The first column is marks of different variants. † denotes the classification loss of a single mask branch.**

| | $\mathcal{L}^{cls}_{Intra}$ | $\mathcal{L}^{reg}_{Intra}$ | $\mathcal{L}^{cls}_{Inter}$ | $\mathcal{L}^{reg}_{Inter}$ | C→F | C→B | S→C |
|---|---|---|---|---|---|---|---|
| A | - | - | - | - | 44.0 | 30.7 | 55.8 |
| B† | ✓ | - | - | - | 47.6 | 36.2 | 58.9 |
| C | ✓ | | | | 48.0 | 36.6 | 59.4 |
| D | ✓ | ✓ | | | 48.9 | 40.7 | 60.3 |
| E | ✓ | | ✓ | ✓ | 48.8 | 40.1 | 61.1 |
| F | ✓ | ✓ | ✓ | | 49.3 | 41.3 | 62.3 |
| G | ✓ | ✓ | | ✓ | 49.5 | 41.5 | 62.1 |
| H | ✓ | ✓ | ✓ | ✓ | 51.1 | 41.8 | 63.8 |

ratio is $1 - r$). We find that when the mask ratio difference between a pair of mask images is too large, the performance of the network will be more unfavorable. Only when $r = 0.5$ can it provide a fairer context consistency reasoning training for Inter/Intra-CCR.

**Different Masking Mode:** Although we find the optimal value of $b$, we wonder if random mask patch sizes are more beneficial to the performance of SOCCER. As illustrated in Figure 4 (a), we conduct two group experiments, one selects random sizes from the entire range (**blue bars**), and another selects random sizes from (24,32,40) (**dark blue bars**), which is empirically validated as the optimal values in Figure 3 (a). The latter is significantly better than the former but still averages 1.2% lower than the fixed size of $b = 32$. We also explore more possibilities for mask image pairs in SCM

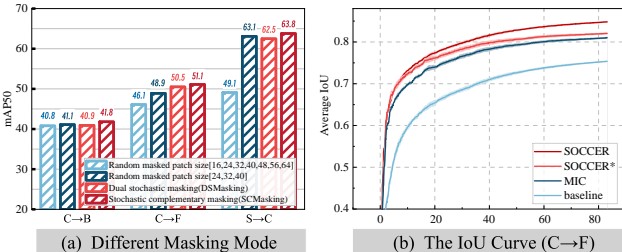

(a) Different Masking Mode  (b) The IoU Curve (C→F)

**Figure 4: Further study of different masking modes and the quality of pseudo labels in our SOCCER.**

module. Specifically, we turn the stochastic complementary masking strategy into a dual stochastic masking strategy (*i.e.*, the masks of $\mathcal{V}_1$ and $\mathcal{V}_2$ are generated respectively). But the result (**red bars**) is still lower than the complementary strategy. This mainly results from that: 1) Random masking brings some overlap of vision clues, so it is easier for the student model to make consistent reasoning based on shortcuts to the same visible regions. 2) Stochastic complementary masking provides greater challenges for student model training, which improves the model's ability to distinguish subtle differences between confusing categories in the target domain.

**Localization Quality of Pseudo Labels:** We further explore the IoU between pseudo labels and ground truth in comparison to baseline[1], MIC, and ours in Figure 4 (b). SOCCER has more accurate localization than other methods. SOCCER* denotes the removal of $\mathcal{L}^{reg}_{Intra}$, which is lower than SOCCER thus explaining the performance difference between *row E* and *row H* in Table 4.

---

[1] Since there is no teacher model in the baseline, for a fair comparison, we take the baseline's output with a confidence threshold $\delta = 0.8$ to filter unreliable predictions.

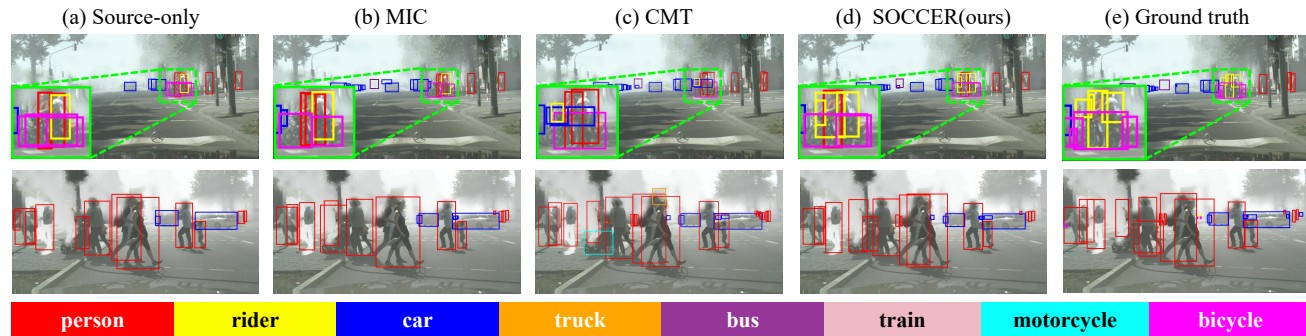

Figure 5: Qualitative results comparison on Cityscapes to Foggy Cityscapes for (a) Source Only [41], SOTA: (b) MIC [23] and (c) CMT [2], (d) Ours, and (e) Ground truth. We provide more comparison results in Appendix.

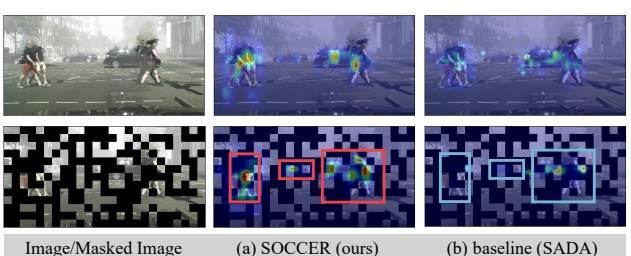

Figure 6: Class-Activation-Map (CAM) visualization. From top to bottom, this is the CAM of SOCCER and baseline for the original image and the masked image.

**Compare with Different Masking Out Augmentation:** Masking out augmentation [5, 11, 61] is widely employed for its effectiveness and efficiency. By deleting parts of regions, networks can learn originally less sensitive but important information thus increasing the perception field. Since the similarity, we further compare them with our SCMasking strategy. Note that we conduct the above masking out augmentation in dual branches of SCM respectively, same with the setting of DSMasking. In Table 5, the single mask region methods Cutout and Random Erasing are unfavorable for context consistency training, reducing 11.1% and 10.6% mAP compared to SCMasking. While GridMask uses structured discard regions, random mask sizes introduce more uncertainty (we prove in Figure 4 (a) that random mask sizes are adverse to performance), and regularly visible regions reduce the initiative of the network to learn context information.

Table 5: Results on Cityscapes → BDD100k with different masking out methods in SCM module. DSMasking denotes the dual stochastic masking and SCMasking denotes our stochastic complementary masking. We compare three methods: Cutout [11], Random Erasing [61], and GridMask [5].

| Selected Method | $mAP_{0.5}$ | $mAP_{0.75}$ | $mAP_{0.5:0.95}$ |
|---|---|---|---|
| Cutout [11] | 30.7 | 11.2 | 12.1 |
| Random Erasing [61] | 31.2 | 11.5 | 14.5 |
| GridMask [5] | 33.5 | 13.9 | 16.2 |
| DSMasking | 40.9 | 18.2 | 20.9 |
| SCMasking | **41.8** | **18.7** | **21.1** |

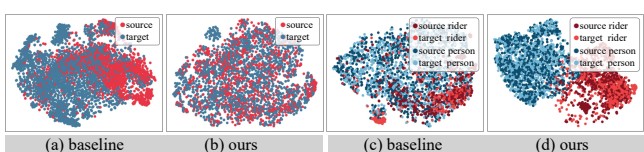

Figure 7: T-SNE visualization. (a), (b) and (c), (d) are domain-level features and category-level features from two domains.

## 5.5 Quantitative Visualization Analysis

**Qualitative Results Visualization:** Figure 5 visualizes some detection results of Source-only [41], MIC [23], CMT [2] and SOCCER, along with the ground truth for comparison. Our SOCCER can accurately localize and classify most objects indicating that Inter-CCR enhances the feature's representation to distinguish confused categories (such as "person" and "rider" in $row1$) and Intra-CCR models the context correlation to detect objects in dense scenes ($row2$).

**CAM Visualization:** We further conduct Class-Activation-Map (CAM) [43] for a visual explanation of how our network uses contextual information to reason. As observed in Figure 6 (a), SOCCER can still use visible contextual clues to reason the occluded target even when the object is occluded (**red boxes**). Comparison Figure 6 (b), the baseline model without masking strategy only focuses on visible areas and lacks context reasoning capability (**blue boxes**).

**T-SNE Visualization:** As illustrated in Figure 7 (a) and (b), our SOCCER aligns the distribution of two domains (**C** & **B**) better than the baseline showing excellent adaptation. Figure 7 (c) and (d) show that SOCCER extracts more discriminative features to distinguish the confusion categories: "rider" and "person" and aligns the same category of two domains well, while baseline mixes them up.

## 6 CONCLUSIONS

In this paper, we propose a stochastic context consistency reasoning (SOCCER) network to model the underlying contextual correlation of the target domain for domain adaptive object detection. Stochastic complementary masking introduces visual diversity and prevents the model from heavily relying on specific visual features. Intra-CCR and Inter-CCR modules model the context correlation of the target domain and enhance the representation ability of extracted features. Experiments on three benchmark cross-domain experiments demonstrate that our network dramatically learns spatial and semantic knowledge in the target domain.

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
