# OpenReview forum: "Stochastic Context Consistency Reasoning for Domain Adaptive Object Detection"
_acmmm.org/ACMMM/2024/Conference — MM2024 Poster_

### Official Review · Reviewer_5QSQ · 2024-05-07

**Rating:** 3
**Confidence:** 4

**Summary:**

This paper explores the significance of mask methods and consistency learning for domain adaptive object detection. It proposes a stochastic context consistency reasoning network to improve domain adaptive performance.

**Strengths:**

This paper is well written, and fully discusses the impact of mask and consistency learning on domain adaptive object detection. The experimentation is comprehensive.

**Limitations:**

Lack of novelty and technical contribution. The paper is more like a simple extension of *MIC: Masked Image Consistency for Context-Enhanced Domain Adaptation*.
As problem raised in the abstract *such category-level consistency supervision suffers from poor quality of pseudo labels*, the proposed methods in this paper do not seem to address such issue.

There are still some detailed concerns:
1. The SOTA results did not include *Harmonious Teacher for Cross-domain Object Detection (CVPR2023)*. Please compare with it.
2. Is the *red bars* in line 796 should be *orange bars*?
3. Doubt the significance of proposed *SCM*. Did the authors try to constructs more inter-changeable views $\{V_1, V_2, V_3, ...\}$ from one target images for *inter-changeable context consistency reasoning*?

**Suitability:**

1

---

### Official Review · Reviewer_Raxc · 2024-05-16

**Rating:** 4
**Confidence:** 2

**Summary:**

The paper titled "Stochastic Context Consistency Reasoning for Domain Adaptive Object Detection" presents a novel approach to improving Domain Adaptive Object Detection (DAOD). DAOD aims to enhance the performance of object detectors on unlabeled target domains by leveraging labeled source domain data. This paper proposes a stochastic context consistency reasoning (SOCCER) network for DAOD, aiming to address the sub-optimal learning issue caused by category-level consistency learning in existing literature. Stochastic complementary masking is introduced to enhance the context correlation ability of the network on the target domain. Inter-CCR and Intra-CCR modules are designed to enable the student network to learn from self-supervised signals that ensure context consistency across different views, as well as from pseudo labels that ensure context consistency with the teacher network that obtains the complete view.

**Strengths:**

1. **Novelty**: The introduction of the Stochastic Complementary Masking (SCM) and the dual context consistency reasoning modules (Inter-CCR and Intra-CCR) are innovative contributions to the field of DAOD. These components address the critical issue of context understanding.
2. **Theoretical Approach**: The paper presents a sound theoretical framework that combines self-training with stochastic masking and context consistency reasoning. This multi-view consistency approach is well-justified and enhances the learning process by providing robust contextual cues.
3. **Technical Correctness**: The proposed method is technically robust, with clear definitions and justifications for the SCM, Inter-CCR, and Intra-CCR modules. The mathematical formulations and algorithmic steps are well-detailed, ensuring reproducibility and understanding.
4. **Adequate Evaluation**: The paper conducts extensive experiments on three DAOD benchmarks, demonstrating significant performance improvements over existing state-of-the-art methods. The results are supported by comprehensive ablation studies that validate the contributions of each module.
5. **Clarity**: The paper is well-structured and clearly written, with detailed explanations of the proposed method, its components, and the experimental setup. Figures and tables are effectively used to illustrate key concepts and results.
6. **Applications**: The proposed SOCCER network has practical implications for various applications, such as autonomous driving, intelligent surveillance, and industrial automation, where robust object detection across different domains is critical.

In summary, this paper presents a satisfactory level of innovation and the quality of the manuscript is very high. The experiment was carefully conducted, and the subsequent analysis was comprehensive, effectively confirming the effectiveness of the proposed method.

**Limitations:**

**No clear explanations of the contributions to multimedia/multimodal research**: The most import thing may be that as a submission for ACM MM, the authors could provide a full discussion on the research significance, prospects, and feasible next steps of the proposed method in the fields of multimedia/multimodal research. This is of significance for promoting related research.

**Suitability:**

2

---

### Official Review · Reviewer_8CnX · 2024-05-20

**Rating:** 4
**Confidence:** 2

**Summary:**

This article presents further research on the task of Domain Adaptive Object Detection (DAOD) by proposing a method based on stochastic mask consistency. It introduces complementary masks and two types of loss calculation to achieve content consistency learning, resulting in state-of-the-art detection performance on three benchmarks. The method provides a more efficient approach for adapting a labeled source domain object detection network to an unlabeled target domain.

**Strengths:**

1. The method of complementary masking (Stochastic Complementary Masking) is interesting. It allows the network to share the same set of labels across different augmented images.

2. The experiments conducted in the paper are comprehensive. In addition to ablation experiments, they compare different types of masks, analyze parameter sensitivity, and provide quantitative visualization analysis.

3. The paper is well-structured and easy to understand.

**Limitations:**

1. The necessity of complementary masks is uncertain. In the experiment of Different Masking Out Augmentation, the results of DSMasking and SCMasking seem to be comparable within the margin of error.It is unclear whether complementary masks have a distinct advantage in terms of contextual connections or content understanding.

2. The advancement of the proposed method. In the experiment from Cityscapes to Foggy Cityscapes, the results of SCM appear to have both advantages and disadvantages compared to MIC and CMT, making it difficult to claim a significant improvement.

3. The novelty of the paper. This is not the first proposal of a mask method, and the Inter-CCR and Intra-CCR modules can be considered somewhat ordinary for this work.

**Suitability:**

2

---

### Official Review · Reviewer_ziFA · 2024-05-24

**Rating:** 3
**Confidence:** 2

**Summary:**

Domain Adaptive Object Detection (DAOD) aims to improve the detector's adaptivity to unlabeled target domains by labeling the source domain. Recent advances have utilized self-training frameworks that enable student models to learn target domain knowledge using pseudo-labels generated by teacher models. A stochastic contextually consistent reasoning (SOCCER) network with a self-training framework is proposed.

**Strengths:**

The overall framework of the paper is good.
The work on the paper is very good and representative of the type.

**Limitations:**

1、Insufficient introduction to the context in the introductory section
2、The structure and logic of the article is not made very clear.

**Suitability:**

2

---

### Meta-Review · Area_Chair_judC · 2024-07-01

**Recommendation:** Accept (Poster)
**Confidence:** 4

**Metareview:**

This paper proposes a novel approach to address key challenges in DAOD by enhancing context correlation and robustness. Reviewers have noted the paper's strong theoretical framework, comprehensive experiments, and practical implications for applications. Although some concerns were raised regarding the technical details; The overall innovation and technical contribution of the proposed method, warrant acceptance. Therefore, I recommend accepting this paper and hope the authors can carefully revise the manuscript according to the reviewers' comments in the camera-ready version.